# Implementation of a screening, brief intervention and referral to treatment programme for risky substance use in South African emergency centres: A mixed methods evaluation study

Claire van der Westhuizen[1]*, Bronwyn Myers[2,3], Megan Malan[1], Tracey Naledi[4,5], Marinda Roelofse[4], Dan J. Stein[6], Sa'ad Lahri[7,8], Katherine Sorsdahl[1]

1 Alan J Flisher Centre for Public Mental Health, Department of Psychiatry and Mental Health, University of Cape Town, Cape Town, South Africa, 2 Alcohol Tobacco and Other Drug Research Unit, South African Medical Research Council, Cape Town, South Africa, 3 Department of Psychiatry and Mental Health, University of Cape Town, Cape Town, South Africa, 4 Western Cape Department of Health, Cape Town, South Africa, 5 School of Public Health Medicine & Family Medicine, University of Cape Town, Cape Town, South Africa, 6 SA MRC Unit on Risk & Resilience in Mental Disorders, Department of Psychiatry & Neuroscience Institute, University of Cape Town, Cape Town, South Africa, 7 Department of Emergency Medicine, Stellenbosch University, Stellenbosch, South Africa, 8 Khayelitsha Hospital Emergency Services, Cape Town, South Africa

* claire.vanderwesthuizen@uct.ac.za

**Data Availability Statement:** The data are owned by the Western Cape Department of Health and

## Abstract

### Background

Screening, brief intervention, and referral to treatment (SBIRT) for risky substance use is infrequently included in routine healthcare in low-resourced settings. A SBIRT programme, adopted by the Western Cape provincial government within an alcohol harm reduction strategy, employed various implementation strategies executed by a diverse team to translate an evidence-based intervention into services at three demonstration sites before broader programme scale-up. This paper evaluates the implementation of this programme delivered by facility-based counsellors in South African emergency centres.

### Method

Guided by the Consolidated Framework for Implementation Research, this mixed methods study evaluated the feasibility, acceptability, appropriateness and adoption of this task-shared SBIRT programme. Quantitative data were extracted from routinely collected health information. Qualitative interviews were conducted with 40 stakeholders in the programme's second year.

### Results

In the first year, 13 136 patients were screened and 4 847 (37%) patients met criteria for risky substance use. Of these patients, 83% received the intervention, indicating

applicants may apply online to the National Health Research Database (https://nhrd.hst.org.za/). The authors confirm they had no special access or privileges that other researchers would not have.

**Funding:** CvdW received research funding via a postdoctoral fellowship award via the DELTAS Africa Initiative [DEL-15-01], African Academy of Sciences (https://www.aasciences.ac.ke/), funded by the Wellcome Trust (https://wellcome.ac.uk/). KS recieved funding from the Western Cape Office of the Premier. The funders had no role in study design, data collection and analysis, decision to publish, or preparation of the manuscript.

**Competing interests:** The authors have declared that no competing interests exist.

programme feasibility. The programme was adopted into routine services and found to be acceptable and appropriate, particularly by stakeholders familiar with the emergency environment. These stakeholders highlighted the burden of substance-related harm in emergency centres and favourable patient responses to SBIRT. However, some stakeholders expressed scepticism of the behaviour change approach and programme compatibility with emergency centre operations. Furthermore, adoption was both facilitated and hampered by a top-down directive from provincial leadership to implement SBIRT, while rapid implementation limited effective engagement with a diverse stakeholder group.

## Conclusion

This is one of the first studies to address SBIRT implementation in low-resourced settings. The results show that SBIRT implementation and adoption was largely successful, and provide valuable insights that should be considered prior to implementation scale-up. Recommendations include ensuring ongoing monitoring and evaluation, and early stakeholder engagement to improve implementation readiness and programme compatibility in the emergency setting.

## Introduction

Globally, low- and middle-income countries (LMICs) have limited resources to prevent the harms associated with risky substance use [1, 2]. As in many LMICs, risky substance use is highly prevalent in South Africa, with 13% of adults meeting criteria for a lifetime substance use disorder and 43% of alcohol users reporting heavy episodic drinking [3, 4]. This modifiable risk factor contributes to South Africa's quadruple burden of disease due to HIV and other infectious diseases, injuries and non-communicable diseases [5]. The World Health Organization (WHO) has supported the scale up of substance use screening, brief intervention, and referral to treatment (SBIRT) within healthcare services as a means of reducing risk of injury and other health consequences associated with risky substance use [6]. Such a SBIRT programme often involves the coordination of input from multiple disciplines and management structures, and may include a continuum of care from pre-screening and screening to brief intervention, brief treatment and referral to treatment for high-risk cases [7].

SBIRT for risky substance use is an innovation which has been tested and implemented in diverse settings [8]. In some high-income countries (HICs), such as the United States, SBIRT has become a routine part of primary and emergency or trauma care [9]. As such the bulk of the evidence has emerged from HICs [10, 11]. Research in LMICs regarding SBIRT is relatively sparse and a few countries dominate this literature, notably Brazil [12, 13], South Africa [14–16] and Thailand [17, 18]. In South Africa specifically, SBIRT has been tested antenatal care; HIV, TB and chronic disease services and emergency centres [15, 19–23]. At the present time, in South Africa, there is ongoing work on establishing the effectiveness of interventions to address substance use [23, 24].

Although there is a substantial evidence base for the efficacy of SBIRT, several questions remain unanswered. First, in certain settings, particularly emergency centres, outcomes are inconsistent [25–27], and SBIRT seems to benefit certain populations more than others [28, 29]. Furthermore, there are gaps in SBIRT evidence for certain populations, such as patients presenting to LMIC emergency centres. Second, there is uncertainty regarding how best to deliver and implement SBIRT, and about which contextual factors may affect SBIRT implementation, particularly given the inconsistent findings [7, 11, 30]. Factors which have been

explored mainly fall under the CFIR 'inner setting' and include provider knowledge, attitudes and behaviours, including stigma related to substance use and behaviour change [31–33]; supervision and support structures for implementation [34]; and leadership for the SBIRT programmes within implementing organisations [35, 36]. However, certain aspects of the implementation context, particularly in the 'outer setting', have been neglected, such as policies and directives in the environment beyond the organisation where an intervention is implemented [30]. Third, there are questions regarding how the use of frameworks, such as the Consolidated Framework for Implementation Research (CFIR) [37], can explain variation in SBIRT implementation outcomes. This approach has not been widely applied [38], as most studies using the CFIR identify barriers and facilitators to implementation, without linking the findings to implementation outcomes, such as adoption or appropriateness [38, 39]. Implementation research is embedded in some HIC healthcare organisations [40]'[41], and is used to inform clinical practice. This is less common in LMICs; although, examples of implementation research being conducted alongside real-world implementation exist, but mainly in primary care [12, 20].

Recently, a unique opportunity arose to address these gaps in the literature by evaluating the implementation of a SBIRT programme, labelled the Teachable Moment programme, in emergency centre settings in South Africa. We employed the CFIR to examine contextual factors, within the implementing organisation and in the outer policy setting, which influenced SBIRT implementation in the low-resourced setting, adding to the sparse LMIC literature on real-world SBIRT service provision. Furthermore, we used our CFIR-based findings to explore the implementation outcomes of the SBIRT programme.

This programme had been tested in a randomised controlled trial (RCT) completed in 2013 in collaboration with the Western Cape Department of Health [15]. The trial, conducted in Western Cape emergency centres, found that the SBIRT programme was effective, acceptable and feasible to implement when task-shared to facility-based counsellors who were added to the emergency care team [15] [42]. The Western Cape Department of Health put forward the SBIRT programme for implementation when the Office of the Premier of the Western Cape province introduced an alcohol harm reduction gamechanger strategy [43], based on WHO's alcohol harm reduction recommendations [44]. The SBIRT programme was adopted as a gamechanger intervention. The existing collaboration with the original research team was the foundation for further collaborative work in implementing the Teachable Moment programme in three emergency centres serving communities with high levels of alcohol-related injuries. Funding was provided by the Premier's office and the Department of Health contracted local non-profit organisations to recruit health counsellors to deliver the programme. These counsellors were trained in July 2016, placed in the emergency services as part of the emergency centre team, and the programme was rolled out in August 2016. The programme involves: (i) screening adult (≥18 years) patients with non-life-threatening injuries or conditions for risky alcohol or drug use utilising the Alcohol, Smoking and Substance Involvement Screening Tool (ASSIST), (ii) providing a brief motivational interviewing-based intervention session for patients reporting recent substance use and ASSIST scores greater than 6 for alcohol or 1 for drugs [45], (iii) offering two further intervention sessions based on problem-solving therapy and (iv) referring people who screened at high risk for substance-related harms (with ASSIST scores above 26) to the regional Department of Social Development for further treatment.

Several implementation strategies (see Fig 1) were employed to facilitate the integration of the programme into usual care from the pre-implementation phase to initial implementation and maintenance of programme operations. Various stakeholders were involved at different stages. The provincial Department of Health and the researchers provided initial support in the pre-implementation and implementation phases, decreasing their involvement in the

**Fig 1. Implementation strategies and stakeholders responsible by implementation phase.**

maintenance phase. The researchers had completely handed over their support roles by the maintenance phase. Through the implementation phases, the regional Department of Social Development and the district Department of Health offices increased their support and by the maintenance phase these offices were driving the SBIRT programme, supported by the non-profit organisations.

The implementation strategies were executed by a diverse implementation team to translate an evidence-based intervention into usual services at three demonstration sites before considering broader scale-up of the programme. The current study aimed to evaluate the first two years of implementation of a SBIRT programme in South African emergency centre settings and identify factors as defined by the CFIR, that influenced specific implementation outcomes, namely feasibility, acceptability, appropriateness and adoption of the evidence-based programme. In LMICs, the available human and financial resources for SBIRT are limited and task-sharing approaches, using lay health workers to deliver interventions, are more feasible in these contexts [46], as compared to some programmes in HICs where nurses, doctors or social workers deliver task-shared interventions [47]. Task-sharing or task-shifting describes the use of non-specialists to deliver services [48]. Evaluations of such low-cost programmes using lay health workers, as in the Teachable Moment programme, are vital to inform future SBIRT efforts in low-resourced settings.

## Methods

Mixed methods were employed to evaluate implementation outcomes and are reported according to accepted guidelines [49]. A sequential explanatory study design was used [50] whereby quantitative routine programme data from the first year of the programme were accessed and analysed, followed by qualitative interviews with stakeholders during the second

**Table 1. Definitions of implementation outcomes.**

| Implementation outcome | Definition* | Operationalised for the study |
|---|---|---|
| Feasibility | "Feasibility is defined as the extent to which a new treatment, or an innovation, can be successfully used or carried out within a given agency or setting." | We operationalised feasibility according to the numbers of patients meeting criteria for risky substance use who received at least the first session of the intervention. In the previous RCT, 20% of eligible patients refused study participation. Since the Teachable Moment programme was implemented in services without compensation for patient time spent, we allowed for 40% refusal. Thus, we operationalised success for this construct as at least 60% of eligible patients receiving at least one session. |
| Acceptability | "Acceptability is the perception among implementation stakeholders that a given treatment, service, practice, or innovation is agreeable, palatable, or satisfactory." | We operationalised acceptability as stakeholder's positive perceptions of aspects of the Teachable Moment intervention, such as the aim of the intervention, the behaviour change approach used, intervention content, the supervision model, the addition of counsellors to the emergency centre, etc. |
| Adoption | "Adoption is defined as the intention, initial decision, or action to try or employ an innovation or evidence-based practice. Adoption also may be referred to as "uptake."" | We operationalised adoption as: (i) stakeholders' commitment to implementing and supporting Teachable Moment programme operations; (ii) stakeholders' integration of the Teachable Moment programme into their organisational structures and services and (iii) cooperation between the stakeholders implementing the programme, evidenced by regular communication, sharing of data and joint action taken. |
| Appropriateness | "Appropriateness is the perceived fit, relevance, or compatibility of the innovation or evidence based practice for a given practice setting, provider, or consumer; and/or perceived fit of the innovation to address a particular issue or problem." | We operationalised appropriateness as: (i) the views of stakeholders on how the Teachable Moment programme fit into usual service operations, particularly acute patient care; (ii) stakeholders' perceptions of patients being open and able to participate in the programme at the acute emergency centre visit and (iii) stakeholders' perceptions of operational priorities and desired allocation of resources. |

*Definitions of implementation outcomes as described in Proctor's taxonomy

year of the programme. In this study, we assessed the feasibility, acceptability, adoption and appropriateness to characterise the implementation of the Teachable Moment programme. Since there is variation in how these implementation outcomes are defined [51], we have used the definitions of feasibility, acceptability, adoption and appropriateness described in Proctor's taxonomy of implementation outcomes [39]. See Table 1 for the definitions used and for the operationalisation of these terms for this study.

Constructs described in the Consolidated Framework for Implementation Research (CFIR) [37] were used to characterise the factors affecting these implementation outcomes and as such were useful in guiding the qualitative data collection, analysis and reporting of the findings. The CFIR was chosen since it was developed as a 'meta-theoretical framework' to conceptualise factors which influence the introduction and sustained use of innovations in health services [37]. Certain CFIR constructs were not suitable for this evaluation and were excluded (see Appendix 1). The use of Proctor's taxonomy of implementation outcomes in conjunction with CFIR constructs is in accordance with implementation research recommendations regarding the use of theoretical frameworks to link factors affecting implementation to observed implementation outcomes [38]. The methods used for the quantitative process evaluation and qualitative components of this study are described separately.

## Quantitative process evaluation

SBIRT programme data were collected by the Department of Health during the first year of the programme, from 1 August 2016 to 31 July 2017. The screening questionnaire administered at

the acute visit included sociodemographics, presenting complaint, number of substance use days in the preceding month and the ASSIST. Process data was collected including numbers of: patients screened, eligible patients and sessions given. Quantitative data were imported into SPSS version 22 and descriptive statistics were conducted. These data were used to assess the feasibility of the Teachable Moment programme.

## Qualitative enquiry of factors affecting implementation

**Participants.** All stakeholders directly involved with the Teachable Moment programme at the provincial, district/regional, non-profit organisation and hospital levels were approached to participate in the study. Two individuals refused participation. A total of 27 individual interviews and 3 focus groups were conducted. Stakeholders were selected based on their involvement with the gamechanger initiative and SBIRT service operations. Those that consented to the interviews included: (i) two provincial Department of Social Development officials who are involved in policy and programme planning for the Department at provincial level; (ii) two provincial Department of Health officials who are involved in policy and programme planning for the Department at provincial level; (iii) three officials from the Department of the Premier involved in the Alcohol Harm Reduction gamechanger who are involved in policy and programme planning for the Premier's Office provincial gamechanger initiatives; (iv) two regional office Department of Social Development officials generally responsible for implementing programmes and providing services at regional level; (v) three district office Department of Health officials generally responsible for implementing programmes at district level; (vi) two hospital managers who oversee all hospital operations and programmes in their respective hospitals; (vii) two EC nursing managers responsible for EC service operations; (viii) two EC medical managers responsible for EC service operations; (ix) six non-profit organisation staff responsible for providing services in healthcare facilities, as contracted by the Department of Health, and (x) three counsellor supervisors employed for the Teachable Moment programme. The Teachable Moment counsellors, who were employed specifically for the SBIRT programme, took part in one of three focus group discussions. The stakeholders interviewed from district and regional offices are programme implementers within the health and social development systems. In these offices, their role could be categorised as that of 'middle managers' in that senior managers initially agreed to the Teachable Moment programme implementation, but then assigned all responsibility for implementation to the district and regional office stakeholders.

**Procedure.** We conducted semi-structured individual interviews with stakeholders in the second year of the programme. These interviews were either conducted at the participants' place of work or at the Centre for Public Mental Health offices, after obtaining written informed consent. Focus groups were conducted with the counsellors at each site. All interviews were audio-recorded and transcribed. Ethical approval was given by the University of Cape Town Human Research Ethics Committee (094/2017), the Western Cape Department of Social Development Provincial Research Ethics Committee (12/1/2/4) and the Western Cape Department of Health Provincial Health Research & Ethics Committee (WC_2017RP39_880), as well as the ethics committees at each of the hospitals.

**Interview schedule.** The interview schedule was developed based on the CFIR, complemented by the authors' knowledge of the emergency centre setting related to: (i) prior experience of conducting the SBIRT randomised controlled trial study in these settings, and (ii) clinical experience in South African emergency centres. Questions included: experiences with the programme and opinions on the intervention, the implementation process, and barriers and facilitators to SBIRT implementation.

## Analysis

Quantitative data were extracted from the routine programme data and imported into SPSS version 25. Descriptive statistics were conducted. Qualitative data were transcribed and imported into NVivo 12 for analysis. The framework approach was utilised [52] and we followed the five steps of (i) familiarisation, (ii) identification of a thematic framework, (iii) indexing, (iv) charting and (v) mapping and interpretation. For the framework, different groups of stakeholders were used as the unit of analysis, for example, district/regional officials or counsellors. We coded the data according to the following implementation outcomes [53]: acceptability, appropriateness and adoption. See Table 1 for the operationalised implementation outcomes. Two researchers coded the transcripts. Any coding disagreements were discussed and resolved. The coding was then compared to calculate the *kappa* statistic and a *kappa* of 0.93 was achieved, indicating high inter-coder reliability.

## Results

The results are described below by implementation outcome. Programme process data is detailed under the feasibility outcome.

### Patient characteristics and feasibility

Over the first year of the programme, 13 136 patients were screened of which 7 163 (55%) were men, and the mean age of the screened patients was 37 years. Of the patients screened, 4 847 patients (37%), met criteria for risky substance use on the ASSIST and were offered the programme, and 3 577 (74%) were male with a mean age of 33 years. Risky alcohol use alone was identified in 3301 patients (68% of eligible patients), risky alcohol and drug use in 893 patients (14%) and risky drug use alone in 653 patients (18%). It proved feasible to deliver the first session at the acute visit with 4 005 (83%) of the 4 847 eligible patients receiving the first session, while only 93 second and third sessions were delivered across the sites.

### Acceptability

Many stakeholders, particularly those more involved in the programme activities such as the counsellors and emergency centre staff, reported finding the Teachable Moment programme acceptable. First, they felt that the programme was meeting a need in communities by addressing an underlying cause of crime, violent injury and other substance-related harms. This was seen as particularly important for emergency centre patients due to the high burden of substance-related injuries in this group, as highlighted by an emergency centre staff member:

> "One of the issues that we have in emergency medicine is we never go to the root cause of why we see so much violence in emergency centres . . . A large percentage of our patients that present to us with resulting trauma, we can predict it will be on a Friday, Saturday, Sunday night . . . we are not addressing those issues. It's like putting a bandage over a gaping wound.." *Hospital staff (participant 27)*

Second, stakeholders' reports of patients' responses to the intervention during their emergency centre visit as well as the changes they described implementing in their lives contributed to stakeholders' positive views of the programme. A non-profit organisation manager reported receiving letters from patients about their good experiences with the programme and counsellors described positive feedback that they received from programme

recipients. The counsellors reported that this positive feedback motivated them in their work and increased their loyalty to the programme. One counsellor recounted a patient's positive response:

> "Patients that is like, 'Oh wow, I did not know that you guys are offering a service here and I have been looking for help. I have got a substance problem and I really want help, and I never knew where to go, but thank you guys for, you know, putting your hand out and wanting to help me.'" *Counsellor (participant 3)*

Third, as reported by all the emergency centre staff interviewed, the programme complemented the emergency centre activities by providing an in-house service which the emergency centre staff began to use for referrals once the programme was integrated into routine services. Additionally, staff were pleased that the programme operations did not interfere with the acute clinical care of the patients; one hospital staff member noted that the programme was "non-invasive" and did not slow them down or add to the workload of the emergency centre staff.

Some stakeholders who were more removed from the programme operations found the programme to be less acceptable, partly due to: (i) their lack of awareness about the evidence base for task-sharing approaches and behaviour change interventions in healthcare settings; (ii) stigma regarding substance users and behaviour change and (iii) concerns about the intervention's transferability to LMICs. Many stakeholders were not familiar with a task-sharing approach, being adamant that a professional person should deliver the intervention, with one provincial official saying, "That is our biggest criticism . . . that social workers should have been considered for the actual intervention." One sceptical stakeholder mentioned, "I respect the WHO [World Health Organization], but do local research . . . does the WHO address patients in [name of local community], noting their social context?" This view was contested, as others were aware of the work that had been done in South Africa, and believed that the evidence provided "a good basis to start from". This resulted in these stakeholders being more accepting of the programme. Positive beliefs about the effectiveness of the behaviour change approach underpinning the intervention were generally held by those stakeholders who were familiar with the emergency centre setting and those who were more involved in the programme delivery, particularly the counsellors. One stakeholder stated that "I don't personally think you can change the behaviour of those people through these sessions". Other stakeholders were also doubtful and these opinions appeared to be based on their own beliefs about substance users being "addicts" and prone to "lying" and promising "that they want to change but tomorrow they are doing the other thing".

Another contentious component of the programme was the weekly supervision of the counsellors. The majority of the stakeholders were satisfied with the supervision noting that it provided an opportunity to discuss and address problems, and allow quality improvement strategies to be implemented. Although the counsellors' opinions on supervision were positive, a few non-profit organisation staff members felt that the supervision sessions were unnecessary or excessive. Some felt that sessions made staff management difficult, with one non-profit organisation manager feeling that the counsellor supervisor treated the counsellors "with kid gloves" allowing their staff to "take advantage and not do what they were supposed to." The manager described difficulties in pushing her staff to reach high targets regarding numbers of patients screened, and felt that the support provided to the counsellors in supervision encouraged excuses from the staff for not reaching their targets.

## Appropriateness

The location of the programme in emergency centres also evoked mixed responses, with hospital staff and counsellors generally endorsing the appropriateness of the programme for this setting. One counsellor reported: "the programme is working perfectly in the EC [emergency centre] because the EC never closes; it is 24/7." A few stakeholders believed that emergency centres are appropriate settings for the programme as counsellors can provide immediate help for those wanting to change their substance use behaviour. Stakeholders highlighted the opportunities to take advantage of the so-called teachable moment when a patient may be considering the negative aspects of excessive substance use.

> "One of the stories I have heard is the fact that somebody had a light bulb moment . . . So they said, "Yoh, Dokter. Ek het nie geweet ek suip so baie nie." [Yoh, Doctor. I didn't know I drink so much.] So while I'm putting stitches in his head, he is like, oh this is a problem. I think this is what we are trying to get. It's pointing out to people that this is not normal." *Hospital staff (participant 27)*

Other stakeholders were sceptical, mainly due to their perceptions of emergency centre patients' physical condition and sobriety at the time of screening and intervention, with one stakeholder believing that the emergency centre patients are "intoxicated . . . not all there". Another stakeholder thought that it would be "very difficult to get real or solid information" from the patients. Additional potential barriers identified by stakeholders regarding the emergency centre programme included patients' concerns around confidentiality in this busy setting, as well as patients' resistance to being screened and receiving an intervention as a result of the long waiting times. This barrier was particularly problematic at one of the sites and is described by a provincial stakeholder: "patients are irritated when they [the counsellors] get to them because they sit for three days, for two days, for long hours . . . If I am sitting there for three days, I do not feel like speaking to a counsellor." However, staff at another site mentioned that the counsellors' activities provided a welcome diversion for some patients which improved the atmosphere in the waiting room and triage area:

> "But most of the patients they like it because it is an icebreaker, because some of them have been sitting there for a while without nothing happening in between. So, we call them . . . the nurses they say we cool them off. Yes. We cool them off." *Counsellor (participant 30)*

Some stakeholders, particularly hospital staff, believed that although the programme operations were acceptable and compatible with the setting, it would be more appropriate to spend resources on other urgent operational priorities, such as staffing, equipment and psychosocial support for the emergency centre staff which should happen "as routinely as you would do a blood pressure". Further priorities directly associated with patient care included improving waiting times, adherence counselling for patients as well as clinical support for psychiatric and self-harm cases seen in the emergency centre. A stakeholder outlined their thoughts on their relative priorities:

> "This is a great intervention, right. . . . but for example like when you struggle; you don't have gloves and then you struggle with basic consumables and you struggle with patients on the floor. I don't know whether the prevention thing relating to alcohol is necessarily something I would invest a lot of my money in when I don't have needles, syringes, gloves, beds. You know if you told me the study is gonna cost R 500 000 and I can maybe get 25 trolleys versus the programme, I would choose the trolleys." *Hospital staff (participant 9)*

## Adoption

One of the main facilitators regarding the adoption of the programme was the top-down directive from the provincial government departments, accompanied by dedicated funding from the Premier's office, to implement the service in the demonstration sites. One emergency centre staff member illustrated this by saying that implementers should use the names of the Head of the Western Cape Department of Health and the Premier and stress that they are "heavily invested in the programme" in order for management to agree to programme implementation. Another stakeholder at provincial level stated that it was easier to approach district offices and hospital management saying, "I am asking you to do this, but I will provide you with resources to be able to do that." Further impetus for programme adoption was provided by the Ministry, with the provincial Minister of Health visiting the sites, causing the hospital staff to be more welcoming of the programme. One of the counsellors mentioned that the hospital staff thought that the programme would fail until "the Minister of Health wants to come; that is when they think 'Wow, they are getting there'."

Despite the fact that a number of stakeholders agreed that such a top-down directive was necessary for programme adoption, this approach functioned as a barrier in some instances. One official mentioned that the initiation of the Teachable Moment did not follow the usual processes, whereby Department of Health district offices would identify a need and request support from the provincial office. For the Teachable Moment programme, the provincial office approached the districts with a need and appropriate intervention, which some felt was starting "the wrong way around". Furthermore, a tight timeline accompanied the top-down directive necessitating a rapid implementation process, leaving some stakeholders feeling as if they were not prepared for implementation and did not have the opportunity to provide input or adapt the process for their setting, particularly those tasked with implementing the services at district/regional and hospital levels. This combination of factors influenced stakeholders' attitudes and their adoption of the programme:

> "Especially the way it was introduced to it was just like, yeah thrown in your lap. . . . There is just a lot of negativity for me around this project. Even if this project could save the world tomorrow, I am not sure that I would buy into it. Because of the negative start that it had, honestly." *District/regional official (participant 1)*

A further barrier to adoption was caused by the incompatibility of the Teachable Moment programme operations with the non-profit organisation model which was functioning within the HIV/AIDS, sexually transmitted infections and tuberculosis services (HAST). The non-profit organisation services use community health workers, who are labelled as HAST counsellors, in usual clinic operating hours on week days, as opposed to the Teachable Moment programme which includes weekend and nigh shifts. The health workers all undergo the same basic training which does not include psychological counselling skills, such as those employed in the Teachable Moment programme. Thus, the Teachable Moment counsellors were seen as being "different" and "not part of the bigger group of counsellors" employed by the organisation. This caused difficulties in that the Teachable Moment counsellors were not able to slot into other sites according to operational needs, and the other health workers could not cover gaps in the Teachable Moment programme roster. Moreover, these health workers involved in usual non-profit organisation services are not provided with the intensive supervision necessary for the implementation of a psychological counselling programme. These differences between the two models caused some difficulties in implementation. Additionally, the non-profit organisations were not familiar with the Teachable Moment model and the rapid

implementation did not allow much consultation with the organisations on this model and only one of the non-profit organisations had worked in mental health services previously. Furthermore, logistical issues proved difficult, such as the compilation of a day and night shift roster.

The involvement of diverse stakeholders from various sectors, and levels within departments, both hindered and facilitated the programme implementation. One of the main barriers to implementation was the diffusion of responsibility that resulted from this intersectoral approach with "too many cooks stirring the pudding". Stakeholders at the district/regional and hospital levels, as well as non-profit organisation stakeholders, reported not taking ownership of the programme since the Department of Health provincial office was driving the implementation, and due to the assistance provided by academic partners. Many stakeholders were confused by the researcher involvement, viewing the programme as just "another research project" and not the implementation of services. This confusion resulted in some stakeholders distancing themselves from the programme implementation. Some did highlight benefits of researcher involvement, including the value added by the researchers to programme monitoring and the facilitation of ongoing feedback loops to improve the programme iteratively.

Stakeholders at provincial and district/regional level also mentioned that this intersectoral approach improved communication between the Departments of Social Development and Health at the district/regional levels, where before the Teachable Moment programme the district/regional offices were "two islands" and the officials "were not able to swim over to one another". In the second year of the programme, this situation had changed significantly:

"You know, for me I can see that Health is now changing their stance or perception or the ownership of it if you know what I mean. Because remember . . . substance abuse falls on Social Development. However, we all know it is a public health matter. So bridging those two together has been a major task for us . . . So, for the first time our clinical setting or the provincial setting will speak to our regional setting–our DSD [Department of Social Development] setting for the first time. So, for me personally, I think that is a great achievement." *Provincial official (participant 33)*

## Discussion

This study contributes to filling the SBIRT evidence gap highlighted in the introduction in the following ways. First, the study is one of the first to examine the implementation of an evidence-based SBIRT programme in an LMIC emergency centre setting and offers recommendations for SBIRT implementation in similar settings. Second, contextual factors influencing implementation were identified in the 'inner setting', such as individuals' SBIRT knowledge and beliefs, as well as in the 'outer setting', such as programme leadership from outside the SBIRT implementation setting. These findings provide a broad perspective of the environment in which healthcare innovations are implemented. Third, the study uses CFIR-based findings to characterise SBIRT implementation outcomes which can inform future implementation and scale-up strategies. The findings are discussed below by implementation outcome.

The Teachable Moment programme proved to be feasible with over 13 000 patients being screened and over 4 000 patients receiving an evidence-based brief intervention in the first year of operation. While delivery of second and third sessions did not prove feasible in the current services, one session may be sufficient for many patients. International programmes integrated into healthcare services frequently include a one-session intervention only, which has been shown to produce positive results and these interventions are included in routine practice recommendations [9, 54].

While some aspects of the implementation plan were successfully completed, other aspects were less successful as reported by stakeholders. Stakeholders' views on factors affecting the acceptability, appropriateness and adoption of the Teachable Moment programme provide insights into programme implementation at provincial, regional/district, hospital and non-profit organisation levels. The factors affecting programme implementation coalesced into three main themes, namely (i) the complexity of stakeholders' individual responses to the programme, which were related to the stakeholders' proximity to the service delivery setting and the compatibility of the programme with the stakeholders' current operational environment, (ii) implementation readiness, specifically related to all stakeholders' access to information regarding the intervention and how to incorporate the programme into services, as well as involvement and commitment of leadership at all levels and (iii) myths and misunderstandings around alcohol use, those who use alcohol and early intervention for behaviour change.

A noticeable trend in stakeholders' views was found in the data according to the stakeholders' proximity to the programme operations and familiarity with the emergency centre environment, which appeared to influence their individual knowledge and beliefs about the intervention and the setting. Hospital emergency centre staff and counsellors were more likely to find the programme acceptable and appropriate for the setting. Those with no experience of the emergency centre environment generally highlighted perceived barriers to the programme, such as patient sobriety at the time of the emergency centre visits, and were less likely to believe that the emergency centre provided opportunities to intervene for risky substance use. Increasingly, in international literature and practice, the emergency setting is being recognised as providing intervention opportunities for a range of risky health behaviours [55]. Common staff barriers to SBIRT implementation internationally include emergency centre staff having limited time to deliver interventions, the need for clinical staff to focus on urgent patient care and the lack of ongoing support for service providers [10, 56], which were addressed by using counsellors added to the emergency centre team for the Teachable Moment programme. Generally, investigators find that hospital staff are open to the provision of SBIRT in emergency centres, but most indicate that another cadre of health worker should be used to deliver the intervention [10, 42]. In the international literature, the emergency centre setting is deemed appropriate by stakeholders, possibly due to the fact that these studies were mainly conducted in high-income countries where SBIRT has been implemented routinely in healthcare settings [9, 47], and the stakeholders interviewed were generally providers who are familiar with these settings, such as trauma surgeons. In the United States, a wide-spread implementation programme shifted focus during the programme, to target high-volume emergency settings for SBIRT scale-up [9].

Additionally, this trend identified in the Teachable Moment programme data, according to the stakeholders' proximity to the programme operations in the emergency centre, appeared to be affected by the perceived compatibility of the Teachable Moment programme with current organisational operations, which influenced stakeholders' openness to the programme. For example, emergency centre staff were accepting of the programme as they realised that emergency centre operations were not affected and that the programme was compatible with their workflows. However, some stakeholders, mainly district/regional and non-profit organisation staff, reported several aspects where the Teachable Moment programme was not compatible with existing workflows and understandings, such as the introduction of the programme by provincial officials, as opposed to the Department of Health districts identifying an operational need. The non-profit organisation staff mentioned barriers to adopting the Teachable Moment programme due to the differences in the requirements of the HAST services, including difficulties in scheduling day and night shifts. Concerns regarding compatibility affect the implementation climate in which a programme is implemented, and the constructs

are related in the CFIR where part of the implementation climate includes "the absorptive capacity for change" and "shared receptivity of involved individuals to an intervention" [37]. The importance of addressing compatibility with existing workflows for programme implementation and sustainability has been highlighted in a number of implementation studies [57–59], accompanied with recommendations to evaluate the context early, before implementation and adapt the interventions and processes for each delivery setting.

In addition to compatibility issues, implementation readiness factors—such as available resources, leadership engagement and access to knowledge and information—proved to be key factors in the Teachable Moment programme implementation. Available financial and human resources, along with the top-down directive from the provincial government, were vital for programme implementation at all three sites. The degree of leadership engagement in programme implementation influenced the process at all levels of leadership, with provincial leadership involvement driving early implementation processes. While senior management in the district/regional offices and hospitals were engaged early, the desired diffusion of information about the programme from leadership to middle management and front-line staff, including the non-profit organisations, did not occur, partly due to the short time frame for implementation. The lack of timely engagement with staff at these levels caused some resistance to adopting the programme from the leadership figures at these levels. Many of these individuals were only engaged when programme implementation was imminent. Recent research has emphasised the importance of including middle management in implementation efforts for a number of reasons, including their role in disseminating information, in supporting employees implementing the intervention to meet their performance targets and in improving the implementation climate [60, 61].

Furthermore, in the Teachable Moment programme, there were gaps in the provision of knowledge on broader concepts underpinning the intervention and implementation process, such as implementing evidence-based practices, as well as the evidence for task-sharing approaches. Implementation studies have found that lack of knowledge regarding interventions is a common barrier [56, 62], yet few studies have highlighted the broader knowledge gaps, such as the evidence for task-sharing approaches. Task-sharing has been employed in a number of different fields [63, 64] and has been used successfully to deliver psychological interventions in LMICs [46, 48], yet is not routine practice in South African healthcare, even though task-sharing is included in the National Mental Health Policy Framework and Strategic Plan 2013–2020 [65].

Teachable Moment programme stakeholders at the provincial, district/regional and non-profit organisation levels expressed their individual knowledge and beliefs regarding alcohol use and its consequences, characteristics of alcohol users, the impact of alcohol use on the health system, as well as effective interventions for alcohol use. Many stakeholders, excluding hospital staff and counsellors, were doubtful that behaviour change approaches were acceptable to patients or effective in reducing risky substance use. These beliefs influenced stakeholders' reactions to the programme, with implications at the outer setting external policy and inner setting operational levels. Similarly, myths and stigma-related beliefs regarding alcohol use and effective interventions are prevalent and have been reported in studies investigating SBIRT programmes. Investigators have found that some stakeholders believe that specialist intervention is required for risky substance use, that behaviour change approaches are unlikely to be effective and that the majority of substance users will be extremely resistant to discussing their substance use and to changing their behaviour [33, 56, 66, 67].

For future scale-up, certain factors should be addressed to facilitate programme implementation. These are outlined below by CFIR construct (see Fig 2). Early and sustained engagement is vital with all levels of staff involved in implementation to address a number of aspects

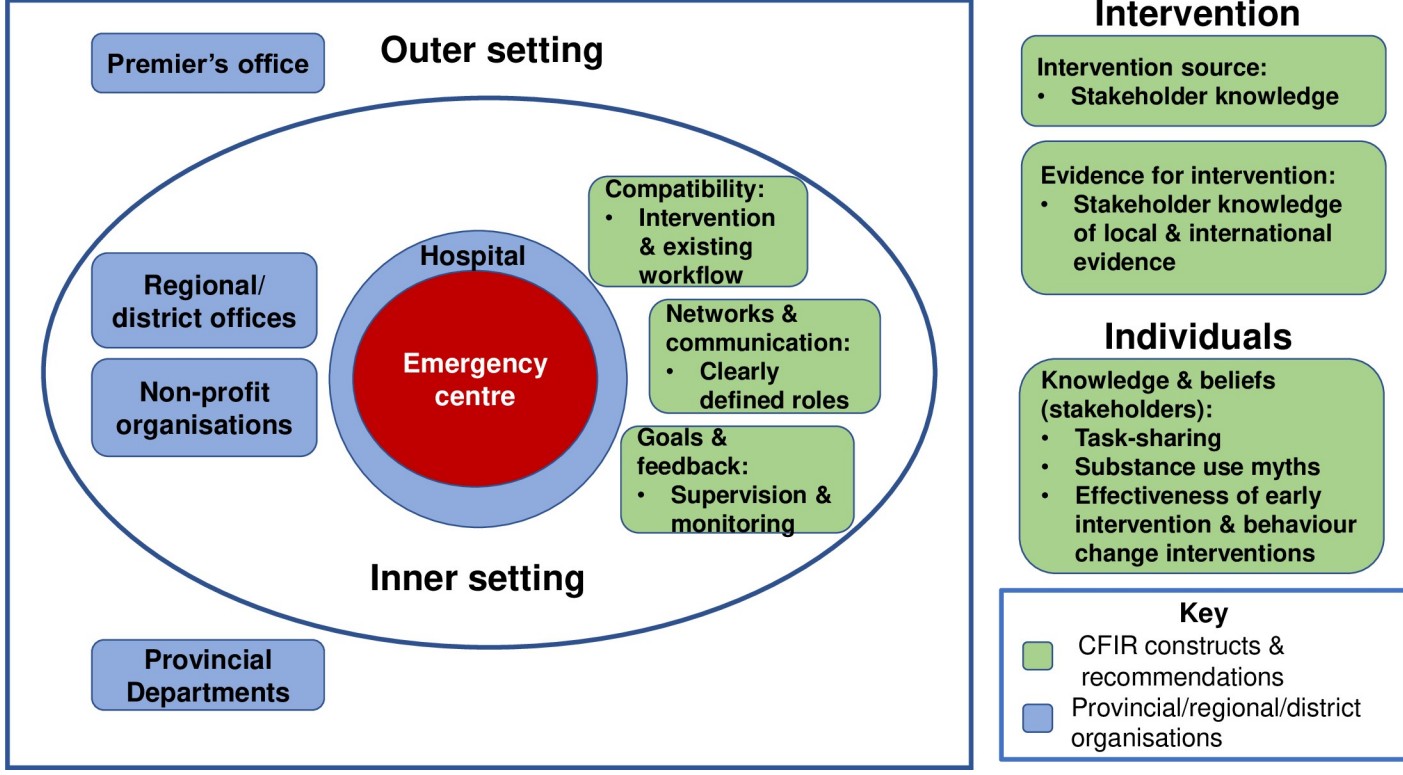

**Fig 2. Recommendations by CFIR construct.**

[60, 62]. This is particularly important for stakeholders who are more removed from the emergency setting, often have more influence on programme implementation and sustainability, and are less familiar with the emergency centre operations First, efforts should be made to address gaps in knowledge regarding the intervention source and evidence base, as well as knowledge and beliefs of individuals regarding broader issues such as knowledge of evidence for task-sharing approaches as well as messaging to debunk myths around substance use and effectiveness of early intervention and behaviour change interventions. Second, existing systems should be mapped in the implementation setting and stakeholders should be engaged on how to address compatibility, in other words to fit the intervention processes into their workflow. This will enable interventions to be adapted as necessary and for organisational operations to be adjusted to accommodate the intervention. Third, with the involvement of a number of different stakeholders, including researchers, which is desirable for implementation research and practice [68], there is a need to clearly delineate roles when establishing the networks and communication in order to avoid confusion in the implementation setting. Fourth, the ongoing supervision and monitoring of goals and providing feedback during implementation was found to be a facilitator of the implementation process and will be vital in implementing the programme in further sites and adapting the programme to suit the specific setting.

## Limitations

Researchers who conducted this evaluation were involved in the previous research, the training of the counsellors and implementation efforts in the healthcare facilities. The team may not have been able to adopt an objective approach to the evaluation. To mitigate this, the team members conducting and analysing the qualitative interviews were not closely involved with

counsellor supervision and day-to-day implementation support of the programme. These team members were aware of the possible bias and gained input on the study findings from a third team member who had not been involved in programme implementation but was familiar with the intervention. Additionally, the mix of implementers and researchers, while desirable in implementation research does add challenges, such as confusion regarding the nature of the programme (research or services). Regular communication was maintained between researchers and implementers during the implementation and efforts to clearly define and revisit roles continued throughout the process. A further limitation of the research is that the qualitative interviews were conducted relatively early in the implementation process, which could have influenced the findings; unfortunately, it was not possible to conduct further interviews. Access to additional data which could have been used to further contextualise the study findings was limited. The data available from the Department of Health comprises total numbers of patients seen in the emergency centre and is not disaggregated by triage code or day of the month. Since the Teachable Moment counsellors can only screen green- and yellow-triaged patients (ie, patients not seriously ill or injured), and the counsellors do not cover night shifts Monday to Thursday, it was not possible to provide the total number of patients available to be screened by the Teachable Moment counsellors. Additionally, data regarding numbers of patients referred on to the Department of Social Development was not available. Patient responses to the Teachable Moment programme will be included in a separate paper due to the volume of data to report.

## Conclusion

The Teachable Moment programme was adopted into routine services and found to be feasible, acceptable and appropriate, particularly by stakeholders familiar with the emergency centre environment, including the counsellors and hospital staff. Other stakeholders not in this position raised a variety of concerns regarding the appropriateness of the programme for the emergency centre setting, the compatibility of the programme with current workflows, influencing programme adoption, as well as the acceptability of the behaviour change approach underpinning the programme. Solutions to the barriers identified mainly involve early engagement regarding stakeholder knowledge and addressing programme compatibility in the implementation setting.

## Supporting information

**S1 File. Use of the CFIR.**
(DOCX)

**S1 Table. CFIR constructs used by domain.**
(DOCX)

**S1 Fig. Outer and inner setting.**
(TIF)

## Acknowledgments

The authors would like to gratefully acknowledge the assistance of the provincial Departments of Health and Social Development, as well as the emergency centre staff at the three hospitals, particularly Dr Aziz Parker. We also acknowledge the facility-based counsellors for giving us their time and for their passion for their work.

## Author Contributions

**Conceptualization:** Claire van der Westhuizen, Bronwyn Myers, Tracey Naledi, Dan J. Stein, Katherine Sorsdahl.

**Data curation:** Claire van der Westhuizen.

**Formal analysis:** Claire van der Westhuizen, Katherine Sorsdahl.

**Funding acquisition:** Claire van der Westhuizen, Katherine Sorsdahl.

**Investigation:** Claire van der Westhuizen, Megan Malan, Tracey Naledi, Marinda Roelofse, Katherine Sorsdahl.

**Methodology:** Claire van der Westhuizen, Bronwyn Myers, Megan Malan, Tracey Naledi, Marinda Roelofse, Katherine Sorsdahl.

**Project administration:** Claire van der Westhuizen, Megan Malan, Marinda Roelofse, Sa'ad Lahri.

**Resources:** Tracey Naledi, Sa'ad Lahri.

**Supervision:** Dan J. Stein, Katherine Sorsdahl.

**Writing – original draft:** Claire van der Westhuizen, Bronwyn Myers, Katherine Sorsdahl.

**Writing – review & editing:** Claire van der Westhuizen, Bronwyn Myers, Megan Malan, Tracey Naledi, Marinda Roelofse, Dan J. Stein, Sa'ad Lahri, Katherine Sorsdahl.

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
