## [Decision Letter · Decision Letter 0]

13 Aug 2019

PONE-D-19-19391

Implementation of a screening, brief intervention and referral to treatment programme for risky substance use in South African emergency centres: a mixed methods evaluation study

PLOS ONE

Dear Dr Claire van der Westhuizen,

Thank you for submitting your manuscript to PLOS ONE. After careful consideration, we feel that it has merit but does not fully meet PLOS ONE’s publication criteria as it currently stands. Therefore, we invite you to submit a revised version of the manuscript that addresses the points raised during the review process.

The reviewers note some important concerns that need to be addressed before your work can make a contribution to the field. Problems are noted regarding your theoretical and methodological approach, description of findings and discussion and conclusion. 

We would appreciate receiving your revised manuscript by Sep 27 2019 11:59PM. To enhance the reproducibility of your results, we recommend that if applicable you deposit your laboratory protocols in protocols.io, where a protocol can be assigned its own identifier (DOI) such that it can be cited independently in the future. For instructions see: http://journals.plos.org/plosone/s/submission-guidelines#loc-laboratory-protocols

We look forward to receiving your revised manuscript.

Kind regards,

Cecilia Benoit

Academic Editor

PLOS ONE

Journal Requirements:

Reviewers' comments:

Reviewer's Responses to Questions

**Comments to the Author**

1. Is the manuscript technically sound, and do the data support the conclusions?

Reviewer #1: No

Reviewer #2: Yes

2. Has the statistical analysis been performed appropriately and rigorously? 

Reviewer #1: N/A

Reviewer #2: N/A

3. Have the authors made all data underlying the findings in their manuscript fully available?

Reviewer #1: Yes

Reviewer #2: No

4. Is the manuscript presented in an intelligible fashion and written in standard English?

Reviewer #1: Yes

Reviewer #2: Yes

5. Review Comments to the Author

Reviewer #1: This study addresses a topic of high relevance to the field. Strengths include the focus on implementation and the LMIC setting. These are important gaps in the current literature. That said, there are some major limitations to this work that appear to restrict its potential contribution to the field. As it is currently written, the theoretical and methodological approach is not sufficiently justified or described, such that the work does not come across as intentional or structured. These limitations may or may not be able to be addressed at this point in time. A strengthened justification for why the study limitations are not fatal flaws is needed to establish the study’s contribution.

1. While I agree that the Introduction (or Methods section) should contain a description of this specific SBIRT programme and the implementation context and strategies, these could be summarized more briefly. More is needed in the Introduction on what is known about SBIRT (including an acknowledgement of the mixed clinical evidence, its role in the broader system/continuum of care, what is known about when and how it is effective) and why studies of implementation are important. The Introduction should clearly outline the rationale for the study and its contribution to the literature. Very little of the vast literature on SBIRT (in different settings/for different substances and levels of use) is cited. There is also no mention of implementation science or how it contributes to system enhancement.

2. The statement of study objectives at the end of the Introduction (lines 113-117) could be strengthened by listing the specific implementation factors and outcomes that were examined (this information is provided later on in the Methods section, but would be good to state up front to better frame the study). The objective(s) should follow from the Introduction and it should be clear how they answer the gap in existing research.

3. More detail is needed to explain how this is a mixed methods study (vs. a multiple methods study; line 119). Using the terminology of Creswell et al. would be helpful to show how the different study components fit together.

4. The study is described as being guided by the CFIR and Proctor’s taxonomy. It is not clearly argued why both are needed, how they fit together, what each brings that complements the other… A clearer framing of the theoretical underpinnings and mixed methods approach (see last comment) would greatly strengthen the front end of the Methods section.

5. Relatedly, a clearer distinction is needed between the constructs of feasibility and adoption. Could the count of patients screened not be considered an indicator of adoption? If possible, the number of patients who were eligible to be screened should be added (e.g., 13,136 patients out of how many were screened?). The meaning of the count of patients screened is hard to interpret in the absence of this information. In addition, if only 1 of 3 planned visits tended to take place, what does that say about feasibility? Finally, no information is provided on the referral to treatment component of SBIRT. This is a critical component of the SBIRT approach and an important aspect of feasibility/adoption. Were there treatment options for those who needed them? Were people referred and did they follow through?

6. Minor point – “game changer strategy” is inconsistently capitalized and written as one/two words (e.g., lines 75 and 142).

7. More information is needed on the sampling strategy for the qualitative component of the study. It looks like efforts were made to recruit stakeholders representing key groups across the system, however, this is not described explicitly. An overall summary of the stakeholder groups and their roles in the system would be helpful (e.g., policy makers, health planners/administrators, clinicians). This is needed to establish how the study answers to its objectives (e.g., who participated in the study and what were they able/not able to speak to?). Currently, the participants section (lines 140-148) is heavy on acronyms and assumes a level of familiarity with the South African system that most readers will not have. A more general statement of stakeholder roles would make this section more widely readable. Finally, is there a justification for the sample size, n=27? Was a sufficient number of people from each (broadly defined) stakeholder group to represent their perspectives?

8. It should be acknowledged as a limitation that patients were not included as participants. This is particularly the case since the Results section refers to “patients’ responses” to the programme and its effectiveness in fostering behavior change (paragraph starting line 208). This form of second-hand reporting (particularly from clinicians involved in delivering the programme) is not a strong approach to evaluating either patient perspectives or their behavior change outcomes. It may not be possible to address this limitation at this point. That said, given that the study is focused on implementation outcomes rather than programme effectiveness, I suggest deleting this paragraph and avoiding any comment on program effectiveness. In the absence of structured evaluation of programme mechanisms, including both positive and negative encounters, this finding is anecdotal.

9. More is needed to justify the claim that programme operations did not interfere with clinical care in the emergency setting (line 223). Was this reported by just one staff member? Did anyone report anything different? Was this explored in a structured fashion?

10. It is not clear what is meant by the quote pertaining to staff taking advantage and not doing what they are supposed to do (line 252). More information is needed on what this finding means and how it relates to issues of staff management (as indicated in line 250).

11. It is noted that there was a lack of compatibility between the SBIRT program and HAST services, and that this caused some difficulties in implementation (line 347). Some specific examples of these difficulties would be helpful here.

12. Many of the findings appear to identify barriers and problems in implementation, raising questions of the extent to which the programme was actually endorsed as appropriate. The authors suggest that those who were more removed from the programme held more negative views of its implementation and impact than those who were closer to the programme. Were the right people asked to report on implementation details? Did all participants know enough about the programme to be able to comment on the details, or are some of them simply echoing negative general perspectives of systems change and/or people who use drugs? Relatedly, the findings indicate a certain level of stigmatizing beliefs held by participants about people who use alcohol and other drugs – rather than stemming from a lack of programme familiarity/proximity per se, this speaks more generally to the negative views that many healthcare providers and administrators hold about problematic substance use. There is a broad literature on occurrence and impact of substance-related stigma in healthcare settings, including emergency room settings, which is relevant to interpreting this finding. As it stands, the relation of these stigmatizing beliefs to programme implementation specifically is not considered in the Results or Discussion sections.

13. There are points made in the Discussion that do not clearly follow from the material presented in the Results section. For example, it is noted that “Available financial and human resources, along with the top-down directive from the provincial government, were vital for programme implementation…” (lines 441-442). How was this assessed? The findings also appeared to identify problems with the top-down directives. Was a structured approach used to assess these features of implementation (e.g., were questions posed to stakeholders about the positive and negative role of these features of implementation and their impact)? Likewise, the Discussion refers to problems in connecting/engaging with middle management, yet this does not clearly follow from the results presented (lines 446-456). How was this evaluated? A thorough review of the Discussion is required to ensure that the interpretation follows from reported findings.

14. Clarification is required on what is meant by “evidence for task-sharing approaches”. Does this pertain to SBIRT interventions or is it about implementation processes more generally? What are the tasks being (or not being) shared?

15. The current Limitations section does not adequately address the limitations of this work and, importantly, how these are expected to affect the findings or what safeguards were used to minimize the impact of potential biases.

Reviewer #2: The authors studied the feasibility and implementability of an SBIRT program in South African emergency rooms throw the examination of relevant outcomes during the first years of the program's implementation, such as screening rates, acceptability, and appropriateness. To do so, they use mainly qualitative but also some quantitative methods. Their results are similar to those encountered in SBIRT implementation studies elsewhere, with particularities relevant to the local context and eventually to other low-middle income contexts. The authors found high levels of stigma towards addiction, represented as a lack of confidence on the effectiveness of the brief interventions, especially among stakeholders not in the clinical sites. They also describe barriers regarding the health network and the clinical workflow. Overall, they found the SBIRT program to be highly implementable in their context, and give some recommendations on how to scale up the innovation.

This study is methodologically sound and addresses one of the more challenging questions in the field of drugs and alcohol brief interventions: how to maximize their implementation to disseminate the strategy. I think it should be accepted with some minor issues that need to be addressed to improve its contextualization and clarity before publication:

MINOR COMMENTS:

Figures:

Figures need to be reviewed and enetually re-made. The implementation process is well explained in figure 1, and the organizational context is depicted in the other figures, but there is some inconsistene and lack of important details:

In Figures S1 and 2: it is not clear why different colors are used. For example, Do they represent hierarchical relationships? Also, the usage of colors does not look consistent between both figures.

Figure 2 looks incomplete. I was expecting to see a summary of the main recommendations for each domain; instead, it only lists the CFIR constructs without any concrete example.

Introduction:

It is a good introduction, but more emphasis could be given to specific aspects of this research regarding the current literature. Other aspects need clarification:

Lines 62 to 69: It is not clear in which aspects the authors expect the implementation to be different due to the socioeconomic background; or if there are clues about that in the new body of literature they mention. I would suggest further illustration.

Line 71: it was difficult for me to follow what program were the authors referring throughout the text: Is the Teachable Moment program the same that was tested in the previous RCT? Is the intervention - training of the counselors implemented here the same that the one used on the RCT program? They mention the 'SBIRT program' or just 'the program' many times, also the 'Game Changer,' but is not clear what program they are referring.

Lines 86 to 88: I understand that the intervention that showed the best effect in the previous RCT was a combined MI + Problem Solving. If that's the case, Why did this program delivered mostly an MI-based intervention? Did this contribute to the supposed lack of evidence ground of the initiative mentioned by some stakeholders?

Methods:

I think this section needs more precision in some critical aspects, particularly more clear operational definitions of the implementation outcomes for this study:

Line 123: Please provide a summary of the CFIR constructs that were not used.

Line 175: the word 'initial' is ambiguous here: does it refers to a general impression or to the idea they had before the program started? It seems to me that the construct of appropriateness was used to assess the suitability some of the innovation's parameters concerning the setting. If this is the case, I think the description given is not clear.

Line 177: it is not clear for me that the Authors mean with 'the intention to try' Later in the paper, they elaborate on the readiness to adopt. Are these concepts equivalent? I would suggest a brief explanation and a more precise operational definition here.

Results:

This part is very clear and consistent in general.

Line 187: Other than meeting criteria for risky substance use, what other requisites were needed to be eligible? Please be precise in the description of the inclusion criteria, because it impacts the overall impression on the program's feasibility the reader will have. Did the ASSIST specific scores define risky substance use?

Line 191: Is it to say that 83% of risky substance users received the first intervention?

Discussion:

The discussion is very well supported by the results, and the paper concludes with recommendations to foster implementation in the future. I think some aspects could be better contextualized or explained to highlight the specific contribution of this research:

Lines 380 to 382: how does this fact relate to local evidence (RCT mentioned in the beginning)?

Lines: 416 to 417: the explanation offered about stake holder's view and how it differs from what's reported in the literature could be further elaborated: it looks like this finding is particularly specific to the context. Also, it is not clear in the last sentence, whether it was a mistake to interview 'distal' stakeholders. Finally, in the recommendations, authors should emphasize a differentiated strategy for early involvement of 'distal' stakeholders based on these findings.

Data: I could not access the dataset; apparently, an application process is needed. I´m not sure whether this precludes from publication in this journal, or if the authors could explain if the dataset is not public for some reason.

6. PLOS authors have the option to publish the peer review history of their article (what does this mean?). If published, this will include your full peer review and any attached files.

Reviewer #1: No

Reviewer #2: Yes: Nicolas Barticevic Lantadilla

---

## [Author Response · Author response to Decision Letter 0]

25 Sep 2019

6 September 2019

Dr Benoit

Academic Editor

PLOS ONE

Dear Dr Benoit

Re: Response to comments for manuscript titled ‘ (PONE-D-19-19391)

Thank you for the helpful comments on this paper. We have addressed them in detail, clarifying the methodology, findings and discussion. Please see our responses below for each point raised.

 Reviewer 1 

1. While I agree that the Introduction (or Methods section) should contain a description of this specific SBIRT programme and the implementation context and strategies, these could be summarized more briefly. More is needed in the Introduction on what is known about SBIRT (including an acknowledgement of the mixed clinical evidence, its role in the broader system/continuum of care, what is known about when and how it is effective) and why studies of implementation are important. The Introduction should clearly outline the rationale for the study and its contribution to the literature. Very little of the vast literature on SBIRT (in different settings/for different substances and levels of use) is cited. There is also no mention of implementation science or how it contributes to system enhancement. 

Thank you for the suggestions for strengthening this section. We cut back the implementation strategies description. We have included more information on research investigating the effectiveness of SBIRT, as well as on SBIRT implementation research (lines 76-95), acknowledging the mixed evidence base. We have also highlighted the contribution that implementation science can make to the field. We have mentioned the SBIRT continuum of care and that contextual factors affect SBIRT implementation and may explain variations in findings (lines 61-63 and 79-87).

2. The statement of study objectives at the end of the Introduction (lines 113-117) could be strengthened by listing the specific implementation factors and outcomes that were examined (this information is provided later on in the Methods section, but would be good to state up front to better frame the study). The objective(s) should follow from the Introduction and it should be clear how they answer the gap in existing research. 

We have added to the sentence describing the study aim, including the implementation outcomes and Consolidated Framework for Implementation Research factors. We have also highlighted the study’s contribution to filling a gap in the literature. See lines 77-101.

3. More detail is needed to explain how this is a mixed methods study (vs. a multiple methods study; line 119). Using the terminology of Creswell et al. would be helpful to show how the different study components fit together. Thank you for this suggestion. We used a sequential explanatory study design, defined according to terminology used by Cresswell (described in Hanson, Cresswell et al, 2004). 

We used sequential quantitative and qualitative study components with findings from the quantitative data informing the qualitative component (see lines 203-206). For example, factors were explored in the qualitative interviews that contributed to the success of the programme in delivering an evidence-based session to over 80% of eligible patients. Additionally, reasons for the low numbers of follow-up sessions were also explored with stakeholders.

4. The study is described as being guided by the CFIR and Proctor’s taxonomy. It is not clearly argued why both are needed, how they fit together, what each brings that complements the other... A clearer framing of the theoretical underpinnings and mixed methods approach (see last comment) would greatly strengthen the front end of the Methods section. 

Thank you for this. We have clarified in the text (see lines 206-233). We followed recommendations found in a systematic review published in Implementation Science on the use of CFIR, where the authors highlight the importance of using CFIR to investigate implementation outcomes, such as those defined by Proctor et al. In assessing the implementation of the study, we used Proctor’s definitions of feasibility, acceptability, adoption and appropriateness. (There is variation in definitions of terms used in implementation research; thus, we decided to use this taxonomy.) The CFIR constructs were used to characterise the factors affecting these implementation outcomes and as such were useful in guiding the data collection, analysis and reporting of the findings. 

5. Relatedly, a clearer distinction is needed between the constructs of feasibility and adoption. Could the count of patients screened not be considered an indicator of adoption? If possible, the number of patients who were eligible to be screened should be added (e.g., 13,136 patients out of how many were screened?). The meaning of the count of patients screened is hard to interpret in the absence of this information. In addition, if only 1 of 3 planned visits tended to take place, what does that say about feasibility? Finally, no information is provided on the referral to treatment component of SBIRT. This is a critical component of the SBIRT approach and an important aspect of feasibility/adoption. Were there treatment options for those who needed them? Were people referred and did they follow through? 

We have added the definitions of each implementation outcome used as described by Proctor et al and further operationalised these for this study (see Table 1). There is some overlap in these terms and the way that they have been used in the literature. We hope that the Proctor definition and our operationalised definition has clarified this. Since the counsellors conducting the screening and delivering the intervention were employed specifically for the Teachable Moment programme, we did not use their activities as indicators of adoption. We would have liked to include the numbers of patients eligible to be screened, however we could not access these data for two reasons. First, the data available from the Department of Health comprises total numbers of patients seen in the emergency centre and is not disaggregated by triage code or day of the month. The majority of these data are captured by emergency centre staff in hard copy triage books. Second, since the Teachable Moment counsellors can only screen green- and yellow-triaged patients, and did not cover week day night shifts, it was not possible to provide these figures. (The Teachable Moment counsellor shifts cover day shifts Monday to Sunday and night shifts Friday to Sunday.) Additionally, data regarding numbers of patients referred on to the Department of Social Development were not available. The referral system underwent a few changes in the first year of the programme. Initially, hard copy referral letters were delivered to the regional Department of Social Development offices. The main problem with this system was that the letters were often lost and the data regarding these referrals were not available from the counsellors or the regional offices. Thank you – we have added these points to the study limitations (see lines 737-746).

6. Minor point – “game changer strategy” is inconsistently capitalized and written as one/two words (e.g., lines 75 and 142). 

Thank you. We have corrected this.

7. More information is needed on the sampling strategy for the qualitative component of the study. It looks like efforts were made to recruit stakeholders representing key groups across the system, however, this is not described explicitly. An overall summary of the stakeholder groups and their roles in the system would be helpful (e.g., policy makers, health planners/administrators, clinicians). This is needed to establish how the study answers to its objectives (e.g., who participated in the study and what were they able/not able to speak to?). Currently, the participants section (lines 140- 148) is heavy on acronyms and assumes a level of familiarity with the South African system that most readers will not have. A more general statement of stakeholder roles would make this section more widely readable. Finally, is there a justification for the sample size, n=27? Was a sufficient number of people from each (broadly defined) stakeholder group to represent their perspectives? 

We invited all stakeholders directly involved with the Teachable Moment programme implementation at the provincial, district/regional, non-profit organisation and hospital levels. We have added this to the text (see lines 250-252). We only had two refusals. Due to the small numbers of people involved, we have not mentioned where these people were employed. We have added the roles for each stakeholder group (see lines 255-290) and hope this clarifies their contribution to the study findings. We have replaced the acronyms DSD, DoH and NPO.

8. It should be acknowledged as a limitation that patients were not included as participants. This is particularly the case since the Results section refers to “patients’ responses” to the programme and its effectiveness in fostering behavior change (paragraph starting line 208). This form of second-hand reporting (particularly from clinicians involved in delivering the programme) is not a strong approach to evaluating either patient perspectives or their behavior change outcomes. It may not be possible to address this limitation at this point. That said, given that the study is focused on implementation outcomes rather than programme effectiveness, I suggest deleting this paragraph and avoiding any comment on program effectiveness. In the absence of structured evaluation of programme mechanisms, including both positive and negative encounters, this finding is anecdotal. 

Thank you. Yes, we believe that patient perspectives on the programme are vital. We do have these data. However, we believe that we already have a great deal of information in this paper and thus decided to write a separate paper on patient perspectives, including data from a small follow-up study of substance use outcomes. We have added this to the limitations. Your opinion on this approach is welcome.

We agree that the second-hand report of patient responses is not an indication of programme effectiveness. We have added to the text to clarify that the stakeholders’ perceptions of patients’ responses contributed to increased acceptability of the programme from the stakeholders’ perspectives. The counsellors were particularly motivated by the reported positive responses. (See lines 360 and 366-367.)

9. More is needed to justify the claim that programme operations did not interfere with clinical care in the emergency setting (line 223). Was this reported by just one staff member? Did anyone report anything different? Was this explored in a structured fashion? 

All the EC staff interviewed reported that the counsellors’ presence was helpful in various ways and that the counsellors had positive interactions with the staff and patients, without hampering clinical care. We have clarified this (see lines 375 and 378). In the qualitative interviews, we asked hospital stakeholders about the positive and negative aspects, specifically exploring patient flow and patient needs in the EC.

10. It is not clear what is meant by the quote pertaining to staff taking advantage and not doing what they are supposed to do (line 252). More information is needed on what this finding means and how it relates to issues of staff management (as indicated in line 250). 

We have added a sentence to clarify: The manager described difficulties in pushing her staff to reach high targets regarding numbers of patients screened, and felt that the support provided to the counsellors in supervision encouraged excuses from the staff for not reaching their targets. (See lines 415-417)

11. It is noted that there was a lack of compatibility between the SBIRT program and HAST services, and that this caused some difficulties in implementation (line 347). Some specific examples of these difficulties would be helpful here. 

We have clarified the differences between the non-profit organisation services and the Teachable Moment programme and added the following sentence: Additionally, the non-profit organisations were not familiar with the Teachable Moment model and the rapid implementation did not allow much consultation with the organisations on this model and only one of the non-profit organisations had worked in mental health services previously. Furthermore, logistical issues proved difficult, such as the compilation of a day and night shift roster (starting line 521).

12. Many of the findings appear to identify barriers and problems in implementation, raising questions of the extent to which the programme was actually endorsed as appropriate. The authors suggest that those who were more removed from the programme held more negative views of its implementation and impact than those who were closer to the programme. Were the right people asked to report on implementation details? Did all participants know enough about the programme to be able to comment on the details, or are some of them simply echoing negative general perspectives of systems change and/or people who use drugs?

Relatedly, the findings indicate a certain level of stigmatizing beliefs held by participants about people who use alcohol and other drugs – rather than stemming from a lack of programme familiarity/proximity per se, this speaks more generally to the negative views that many healthcare providers and administrators hold about problematic substance use. There is a broad literature on occurrence and impact of substance- related stigma in healthcare settings, including emergency room settings, which is relevant to interpreting this finding. As it stands, the relation of these stigmatizing beliefs to programme implementation specifically is not considered in the Results or Discussion sections. 

There were a wide range of opinions expressed regarding the appropriateness of the programme for the emergency centre setting. As the reviewer mentioned, we highlighted that the hospital staff and counsellors who were closest to the programme operations were more likely to report that the programme was appropriate for the setting. As mentioned in response to the question above on sampling strategy, we approached all stakeholders directly involved with the Teachable Moment programme and only 2 people refused so we did have the right group in that sense. Since programme implementation indicators were included in the performance objectives for all stakeholders, they should have had sufficient knowledge of the programme. Many of the stakeholders were required to report on the programme regularly to their superiors.

Regarding stigma related to substance use, we have highlighted a certain aspect related to perceptions that substance users will be resistant to changing their behaviour. We have added specific mention of this in the introduction, results and discussion sections. As the reviewer mentions this belief is prevalent, also among emergency centre staff and studies addressing this are referenced in the discussion.

13. There are points made in the Discussion that do not clearly follow from the material presented in the Results section. For example, it is noted that “Available financial and human resources, along with the top-down directive from the provincial government, were vital for programme implementation...” (lines 441-442). How was this assessed? The findings also appeared to identify problems with the top-down directives. Was a structured approach used to assess these features of implementation (e.g., were questions posed to stakeholders about the positive and negative role of these features of implementation and their impact)? Likewise, the Discussion refers to problems in connecting/engaging with middle management, yet this does not clearly follow from the results presented (lines 446-456). How was this evaluated? A thorough review of the Discussion is required to ensure that the interpretation follows from reported findings. 

We have reviewed the discussion thoroughly. Regarding available financial and human resources, stakeholders did mention that without the addition of resources to the emergency services, the programme may not have been implemented. We mention this in the results: One of the main facilitators regarding the adoption of the programme was the top-down directive from the provincial government departments, accompanied by dedicated funding from the Premier’s office … 

We have quotes underpinning this statement (see below); we were concerned about making the results section too long. We have added a portion of the provincial level stakeholder quote to the results section. The stakeholders were specifically asked about the barriers and facilitators to implementing the Teachable Moment programme.

Provincial official: “But I think the thing that was good is when I was going to the districts and to the facilities to ask them to do this, I was bringing resources – additional resources. That was easier. It made my life a little bit easier. To say I am asking you to do this but I will provide you with resources to be able to do that. So that was a little bit easier.”

Extra quote not added to the results:

Hospital staff member: “what made it easier for us was… So it wasn’t the money taken away from - it was additional money effectively. So, it wasn’t money taken away from one of the already very lean things that we are running here”

Thank you for this. We think this is due to our lack of clarity regarding senior and middle managers. We have clarified the organisational structure under participants with the following: ‘The stakeholders interviewed from district and regional offices are programme implementers within the health and social development systems. In these offices, their role could be categorised as that of ‘middle managers’ in that senior managers initially agreed to the Teachable Moment programme implementation, but then assigned all responsibility for implementation to the district and regional office stakeholders.’ In the results section the opinions of the middle managers on the top-down directive are addressed under adoption. They felt that the Teachable Moment programme was introduced the “wrong way round”, that they weren’t given opportunity to provide any input on the plans and that the added responsibility was just “thrown in your lap”.

We appreciate the comments about the discussion. We have reviewed this section carefully. In some places, we have clarified statements in order to link these more clearly to the results. In one instance, we added a sentence and quote to the results (lines 387-391). This sentence had been omitted from the results by mistake.

14. Clarification is required on what is meant by “evidence for task-sharing approaches”. Does this pertain to SBIRT interventions or is it about implementation processes more generally? What are the tasks being (or not being) shared? 

 We have added information on task-sharing approaches to the introduction (lines 194-198). Task-sharing or task-shifting describes the use of non-specialists to deliver services. Thus in the case of task-shared mental health interventions, these services would not be delivered by psychiatrists or psychologists, but by non-specialist doctors, nurses, social workers, lay health workers etc.

15. The current Limitations section does not adequately address the limitations of this work and, importantly, how these are expected to affect the findings or what safeguards were used to minimize the impact of potential biases. 

We have added text to this section to describe what was done to mitigate the limitations described (725-730 and 732-735).

 Reviewer 2 

1. Figures need to be reviewed and eventually re-made. The implementation process is well explained in figure 1, and the organizational context is depicted in the other figures, but there is some inconsistency and lack of important details: In Figures S1 and 2: it is not clear why different colors are used. For example, Do they represent hierarchical relationships? Also, the usage of colors does not look consistent between both figures. Figure 2 looks incomplete. I was expecting to see a summary of the main recommendations for each domain; instead, it only lists the CFIR constructs without any concrete example. 

Thank you for the suggestions to improve the figures. The colours in Figure 2 and Figure S1 were not used consistently in our original submission. We have changed the colour of the textboxes in Figure S1 to blue for all provincial/regional/district offices; this is consistent with the use of colours in Figure 2. The colours in this figure are not meant to represent hierarchical relationships, thus all these textboxes are now the same colour.

We have added the recommendations provided in the text to Figure 2 under the relevant CFIR constructs.

2. Introduction:

It is a good introduction, but more emphasis could be given to specific aspects of this research regarding the current literature. 

 We have added information on evidence related to SBIRT effectiveness and implementation, as well as gaps in the SBIRT literature (see lines 76-97).

3. Other aspects need clarification:

Lines 62 to 69: It is not clear in which aspects the authors expect the implementation to be different due to the socioeconomic background; or if there are clues about that in the new body of literature they mention. I would suggest further illustration.

We have added a description of how the use of implementation research differs in high- and low- and middle-income countries and added three sentences on how evaluation of task-sharing approaches may add to the literature (94-97 and 194-200).

4. Line 71: it was difficult for me to follow what program were the authors referring throughout the text: Is the Teachable Moment program the same that was tested in the previous RCT? Is the intervention - training of the counselors implemented here the same that the one used on the RCT program? They mention the 'SBIRT program' or just 'the program' many times, also the 'Game Changer,' but is not clear what program they are referring.

We have clarified the terminology used to refer to the RCT and the programme implemented (Teachable Moment) in the text. The programme was tested in the RCT and then implemented as the Teachable Moment programme as part of a province-wide Game Changer initiative addressing alcohol harm reduction. The Teachable Moment programme was implemented using the same components of the programme tested in the RCT, namely: (i) screening processes to identify patients using substances at risky levels, (ii) intervention, (iii) cadre of workers as counsellors, (iv) counsellor training and (v) clinical supervision and support structure.

5. Lines 86 to 88: I understand that the intervention that showed the best effect in the previous RCT was a combined MI + Problem Solving. If that's the case, Why did this program delivered mostly an MI-based intervention? Did this contribute to the supposed lack of evidence ground of the initiative mentioned by some stakeholders? 

Yes, that’s true. The best effect in the RCT was found for the MI + problem-solving therapy (PST) but the group receiving MI alone also improved – regarding substance use scores. The Teachable Moment programme planned to deliver 2 sessions of PST in addition to the first session of MI. Unfortunately, this did not prove feasible in the services. In the RCT, participants received supermarket vouchers in compensation for their time and the RCT counsellors had more time to telephone participants and remind them of their appointments. This did not contribute to the stakeholders’ perceptions of the lack of evidence, as those who mentioned this were not aware of the RCT at all, or any other evidence from South Africa.

6. Methods:

I think this section needs more precision in some critical aspects, particularly more clear operational definitions of the implementation outcomes for this study.

 We have added the definitions of each implementation outcome used as described by Proctor et al and further operationalised these for this study. See Table 1.

7. Line 123: Please provide a summary of the CFIR constructs that were not used.

The following were not included: under the ‘intervention characteristics’ domain, the construct ‘Relative advantage’; under the ‘outer setting’ domain, construct ‘cosmopolitanism’; under the ‘inner setting’ domain, the constructs ‘structural characteristics’ and ‘culture’; under the ‘characteristics of individuals’ domain, the constructs ‘self-efficacy’ and ‘individual stage of change’ and under the ‘process of implementation’ domain, the constructs ‘opinion leaders’, ‘Champions’ and ‘external change agents’ – see File S1.

8. Line 175: the word 'initial' is ambiguous here: does it refers to a general impression or to the idea they had before the program started? It seems to me that the construct of appropriateness was used to assess the suitability some of the innovation's parameters concerning the setting. If this is the case, I think the description given is not clear.

 We have deleted the word ‘initial’ to clarify. Yes, we were using the construct to assess stakeholders’ views of the suitability of the Teachable Moment programme for the setting. (See Table 1.)

9. Line 177: it is not clear for me that the Authors mean with 'the intention to try' Later in the paper, they elaborate on the readiness to adopt. Are these concepts equivalent? I would suggest a brief explanation and a more precise operational definition here. 

 Thank you. As mentioned above, we have addressed this in Table 1.

10. Results:

This part is very clear and consistent in general.

Line 187: Other than meeting criteria for risky substance use, what other requisites were needed to be eligible? Please be precise in the description of the inclusion criteria, because it impacts the overall impression on the program's feasibility the reader will have. Did the ASSIST specific scores define risky substance use?

Thank you, we have clarified this (see lines 136-138). Risky substance use as defined by the ASSIST scores for each substance was used to include patients in the programme.

11. Line 191: Is it to say that 83% of risky substance users received the first intervention? 

Yes, that is correct. 83% of patients identified as risky substance users received the first session at the acute emergency centre visit. 

12. Discussion:

The discussion is very well supported by the results, and the paper concludes with recommendations to foster implementation in the future. I think some aspects could be better contextualized or explained to highlight the specific contribution of this research:

Lines 380 to 382: how does this fact relate to local evidence (RCT mentioned in the beginning)?

We mentioned that it was not feasible to deliver the second and third sessions as part of usual services with the model implemented for the Teachable Moment programme (also see response to reviewer 2, point 5 above). This was not the case in the RCT, where only 20% of participants did not return for further sessions. (The study participants received compensation for their time, in the form of supermarket vouchers for completing assessments). For relation to the RCT, please see response to your comment (no 5) above. It is known that research doesn’t always translate perfectly into implementation; this highlights the need for an implementation focus in effectiveness trials.

13. Lines: 416 to 417: the explanation offered about stake holder's view and how it differs from what's reported in the literature could be further elaborated: it looks like this finding is particularly specific to the context. Also, it is not clear in the last sentence, whether it was a mistake to interview 'distal' stakeholders. Finally, in the recommendations, authors should emphasize a differentiated strategy for early involvement of 'distal' stakeholders based on these findings. 

Thank you. We have added to the text (see lines 620-622). We then elaborate further in the following paragraph. As the reviewer mentions, early involvement of distal stakeholders is important. This is vital to address, given their influence on programming and the fact that they are less familiar with the emergency centre setting. We have added to the discussion to highlight this (see lines 702-704).

14. Data: I could not access the dataset; apparently, an application process is needed. I ´m not sure whether this precludes from publication in this journal, or if the authors could explain if the dataset is not public for some reason. 

Yes, there is an application process as these data are owned by the National Department of Health in South Africa. Any party wanting to access data needs to apply on the National Health Research Database (https://nhrd.hst.org.za/).

We believe that the comments provided have helped us strengthen the paper and we really appreciate the careful reviews. We look forward to hearing from you.

Yours sincerely

Claire van der Westhuizen

---

## [Decision Letter · Decision Letter 1]

25 Oct 2019

Implementation of a screening, brief intervention and referral to treatment programme for risky substance use in South African emergency centres: a mixed methods evaluation study

PONE-D-19-19391R1

Dear Dr. van der Westhuizen,

We are pleased to inform you that your manuscript has been judged scientifically suitable for publication and will be formally accepted for publication once it complies with all outstanding technical requirements.

With kind regards,

Cecilia Benoit

Academic Editor

PLOS ONE

Reviewers' comments:

Reviewer's Responses to Questions

**Comments to the Author**

1. If the authors have adequately addressed your comments raised in a previous round of review and you feel that this manuscript is now acceptable for publication, you may indicate that here to bypass the “Comments to the Author” section, enter your conflict of interest statement in the “Confidential to Editor” section, and submit your "Accept" recommendation.

Reviewer #1: All comments have been addressed

Reviewer #2: All comments have been addressed

2. Is the manuscript technically sound, and do the data support the conclusions?

Reviewer #1: (No Response)

Reviewer #2: Yes

3. Has the statistical analysis been performed appropriately and rigorously? 

Reviewer #1: (No Response)

Reviewer #2: N/A

4. Have the authors made all data underlying the findings in their manuscript fully available?

Reviewer #1: (No Response)

Reviewer #2: Yes

5. Is the manuscript presented in an intelligible fashion and written in standard English?

Reviewer #1: (No Response)

Reviewer #2: Yes

6. Review Comments to the Author

Reviewer #1: (No Response)

Reviewer #2: The authors addressed all the concerns I had with the first version of this manuscript. This new version is much more precise and ready to be published. I have no further comments.

7. PLOS authors have the option to publish the peer review history of their article (what does this mean?). If published, this will include your full peer review and any attached files.

Reviewer #1: No

Reviewer #2: Yes: Nicolas Barticevic Lantadilla

---

## [Editor Report · Acceptance letter]

8 Nov 2019

PONE-D-19-19391R1 

Implementation of a screening, brief intervention and referral to treatment programme for risky substance use in South African emergency centres: a mixed methods evaluation study 

Dear Dr. van der Westhuizen:

I am pleased to inform you that your manuscript has been deemed suitable for publication in PLOS ONE. Congratulations! Your manuscript is now with our production department. 

With kind regards,

on behalf of

Dr. Cecilia Benoit 

Academic Editor

PLOS ONE